# Thioredoxin Reductase-1 as a Potential Biomarker in Fibroblast-Associated HCT116 Cancer Cell Progression and Dissemination in a Zebrafish Model

**DOI:** 10.3390/cancers15010056

**Published:** 2022-12-22

**Authors:** Tharathip Muangthong, Pornnapat Chusangnin, Artchaya Hassametto, Rataya Tanomrat, Prasit Suwannalert

**Affiliations:** 1Department of Pathobiology, Faculty of Science, Mahidol University, Bangkok 10400, Thailand; 2Pathobiology Information and Learning Center, Department of Pathobiology, Faculty of Science, Mahidol University, Bangkok 10400, Thailand

**Keywords:** thioredoxin reductase-1, colorectal cancer, fibroblast, cancer progression, zebrafish, EMT process, oxidative stress

## Abstract

**Simple Summary:**

Prognostic biomarkers, which are used to monitor colorectal cancer status and guide treatment decisions, are crucial. This study aimed to investigate thioredoxin reductase-1 (TrxR-1) expression, which has been related to disease progression in various cancers, for an indication of fibroblast-inducing colorectal cancer progression and metastasis. We suggest thioredoxin reductase-1 as a potential biomarker that can indicate the high proliferative fibroblast-induced-aggressiveness of HCT116 colorectal cancer cells in vitro and in vivo in adult zebrafish models. Therefore, TrxR-1 could be applied as a biomarker for colorectal cancer progression and prognostic evaluation.

**Abstract:**

The tumor microenvironment, especially that of fibroblasts, strongly promotes colorectal cancer (CRC) progression. Progressive cancers usually accumulate high reactive oxygen species (ROS), leading to oxidative stress. The stress relates to the expression of thioredoxin reductase-1 (TrxR-1), which is an oxidative stress sensitivity molecule. This study aimed to investigate TrxR-1 expression as an indication of colon-fibroblast-inducing colorectal cancer progression and metastasis. We found that the high proliferative fibroblast-cultured media (FCM) contained pro-inflammatory cytokines that have a high ability to influence HCT116 and CRC cell progression, when compared with complete media (CM) as a control in terms of growth (CM = 100.00%, FCM = 165.96%), migration (CM = 32.22%, FCM = 83.07%), invasion (CM = 130 cells/field, FCM = 449 cells/field), and EMT transformation while decreasing E-cadherin expression (CM = 1.00, FCM = 0.69) and shape factor (CM = 0.94, FCM = 0.61). In addition, the overexpression of TrxR-1 is associated with cellular oxidant enchantment in FCM-treated cells. A dot plot analysis showed a strong relation between the EMT process and the overexpression of TrxR-1 in FCM-treated cells (CM = 13/100 cells, FCM = 45/100 cells). The cancer transplantation of the adult zebrafish model illustrated a significantly higher number of microtumors in FCM-treated cells (CM = 4.33 ± 1.51/HPF, FCM = 25.00 ± 13.18/HPF) disseminated in the intraperitoneal cavity with TrxR-1 positive cells. The overexpression of TrxR-1 indicated fibroblast-associated CRC progression in HCT116 cells and the zebrafish model. Therefore, TrxR-1 could be applied as a novel biomarker for colorectal cancer progression and prognostic evaluation.

## 1. Introduction

Colorectal cancer (CRC) is a malignant tumor in the large intestine, including the colon and rectum. The mortality rate of this disease was the third highest of any cancer in 2021, and a hospital report found that CRC patients diagnosed at a late stage had a 5-year relative survival rate lower than early-stage patients. [1]. In the last stage or metastasis stage of CRC, only 12% of patients survive more than five years after treatment [2]. Cancer metastasis is the dissemination of cancer cells from the origin site to other organs in the body. This situation is the most severe and remains difficult to cure, causing a significant increase in the mortality rate [3,4]. However, the prognostic evaluation of patients with metastatic CRC remains poor. Therefore, developing novel prognostic and predictive biomarkers, which can be used to monitor disease status and guide treatment decisions, is crucial [5].

The inflammation of the colon has been recognized as a significant factor inducing cancer [6]. In addition, colonic fibrosis following colitis has a strong interaction with colorectal cancer in dysplasia and metastasis. [7]. During fibrosis formation, the highly proliferative fibroblast has an increased ability to secrete pro-inflammatory cytokines, activating the repairing process that has a side effect on cancer progression and metastasis. [8,9]. In cancer progression, the cellular oxidant usually increases, which is associated with enhanced mitochondria activity and supports the energy for cell growth. This results in a rise in reactive oxygen species (ROS), the byproduct of the metabolite process [10,11]. ROS have been widely acknowledged as vital in regulating cancer cell function, including cell proliferation, cell mobility, cell invasion, and activation of the oxidative stress control system [12].

The thioredoxin system is a redox-regulating system that involves many intracellular and extracellular processes, including cell proliferation, gene expression and signal transduction regulation, protection against oxidative stress, and regulation of the redox state of the extracellular microenvironment [13,14]. The thioredoxin system mainly consists of thioredoxin (Trx), thioredoxin reductase (TrxR), and NADPH. Thioredoxin reductase-1 (TrxR-1) plays a major role in reducing thioredoxin with NADPH substrate located in the cytoplasm [13]. In cancer research, TrxR-1 overexpresses in various cancers, including colon cancer, and plays an important role in regulating cancer cell growth by modulating the DNA binding activity of transcription factors [15,16]. Moreover, the expression of TrxR-1 has been associated with drug resistance, cancer migration, and cancer survival [16,17,18]. Therefore, the role of TrxR-1 in CRC invasion and metastasis has been interesting for investigation as a biomarker of progressive CRC.

This study aimed to investigate the sensitivity of an oxidative stress marker, TrxR-1, for the highly proliferative fibroblast-induced HCT116 colorectal cancer cells, and their progression and dissemination in the adult zebrafish model. The positive result for TrxR-1 in cancer progression—growth, migration, invasion, and metastasis—means that it may be used as a potential biomarker to indicate CRC aggressiveness.

## 2. Materials and Methods

### 2.1. Cell Culture and Fibroblast-Cultured Medium Preparation

HCT116 (ATCC^®^ CCL-247TM), colorectal cancer cells, were routinely cultured in completed McCoy’s 5A medium (Sigma-Aldrich, St. Louis, MO, USA). CCD-18Co (ATCC^®^CRL-1459TM), colon fibroblast cells, were cultured in complete Eagle’s minimum essential medium (EMEM) (Sigma-Aldrich, USA). The media were supplemented with 10% heat-inactivated fetal bovine serum (FBS) (Himedia, Maharashtra, India), 1% non-essential amino acids (Capricorn scientific, Ebsdorfergrund, Germany), 1% L-glutamine (Capricorn scientific, Germany), 1% penicillin-streptomycin (Capricorn scientific, Germany), and 2.2 g/L sodium bicarbonate (Sigma-Aldrich, USA). Cells were cultured at 37 °C in a 5% CO_2_ humidified incubator. For fibroblast-cultured media (FCM) preparation, CCD-18co cells were cultured until 80% confluence. Then, the cells were washed with PBS to remove old media and cultured with complete McCoy’s 5A medium for 48 h. The cultured medium was filtered with a 0.22 mm pore size filter to remove debris protein and kept at −20 °C. Fibroblast-cultured media (FCM) were prepared by diluting the cultured media with complete McCoy’s 5A medium (1:4) to perform the experiments.

### 2.2. Cytokine Array

A Human Cytokine Antibody Array (Abcam, Cambridge, UK, Ab133996) was performed with CM (as a control) and FCM following the protocol instructions. The positive cytokines were detected, and the intensities were analyzed with the ChemiDoc imaging system (Bio-Rad Finland, Helsinki, Finland).

### 2.3. Cell Viability by MTT Assay

HCT116 (5 × 10^4^ cells/well) were seeded in 96-well plates and incubated at 37 °C with 5% CO_2_ for 24 h. The cells were treated with various conditions and incubated for 48 h. The media were removed, and 500 μg/mL Thiazolyl blue tetrazolium bromide (MTT) (Panreac AppliChem, Germany) solution was added for incubation at 37 °C with 5% CO_2_ for 3 h. Formazan crystals were dissolved with 100 μL of DMSO (Merck, Darmstadt, Germany). The absorbances were detected with a microplate reader at 570 nm [19].

### 2.4. Wound-Healing Assay

HCT116 were seeded in 6-well plates and incubated at 37 °C with 5% CO_2_ until 100% confluent. The proliferation of the cancer cells was inhibited by pretreatment with 10 μg/mL of mitomycin C (Sigma-Aldrich, USA) for 2 h. A scratched wound was made with a micro-pipette tip and washed to remove the detached cells with phosphate-buffered saline (PBS). The cells were treated with various conditions and incubated at 37 °C with 5% CO_2_. The migration areas were captured at 0, 24, 48, and 72 h after treatment. Then, the migration areas were measured by the Image J program [20].

### 2.5. Boyden Chamber Assay

Transwells (8.0 μm pores size, Corning, Glendale, AZ, USA) were coated with 50 μL of extracellular matrix (ECM) gel from Engelbreth-Holm-Swarm murine sarcoma (Sigma-Aldrich, USA). HCT116 (1 × 10^5^ cells/well) were seeded into coated Transwells and incubated for 24 h. The cells were treated with various conditions, including a control (non-treated cells), conditioned media, and 20 μM H_2_O_2_ as a positive control and incubated at 37 °C with 5% CO_2_ for 72 h. The invaded cancer cells were fixed with methanol and stained with Hematoxylin and Eosin staining (H&E). The result was analyzed by counting the invaded cells under a light microscope for six fields of triplicate [21].

### 2.6. Indirect Immunofluorescence Assay

HCT116 (5 × 10^4^ cells/well) were seeded on sterilized coverslips contained in six-well plates, which were incubated at 37 °C with 5% CO_2_ for 24 h. The cancer cells were treated with various conditions, including a control (non-treated cells), conditioned media, and 20 μM H_2_O_2_ as a positive control, and incubated at 37 °C with 5% CO_2_ for 48 h. The treated cells were fixed with 4% paraformaldehyde and washed with PBS. The cells were permeabilized with 0.25% Triton X-100 (Sigma-Aldrich, USA), and then primary antibody (mouse anti-human E-cadherin (1:1000) and rabbit anti-human TrxR-1 (1:1000) (Abcam, USA)) were added. The cells were incubated at 4 °C overnight, and they were then washed with PBS in triplicate. The secondary antibody (goat anti-mouse Alexa488, goat anti-rabbit Alexa488 (1:1000) (Abcam, USA)) was added and incubated for 90 min at room temperature. The protein expression was captured under a fluorescence microscope [22].

### 2.7. Morphology Change

Morphology changes were determined with a round or polygonal shape instead of the elongated shape of a target cell. HCT116 (5 × 10^4^ cells/well) were seeded in six-well plates and incubated at 37 °C with 5% CO_2_ for 48 h. The media were removed, and various conditions were established, including the control (non-treated cells), conditioned media, and 20 μM H_2_O_2_ as a positive control, and they were incubated at 37 °C and 5% CO_2_ for 48 h. The cells were captured under a light microscope at 400×. A total of 100 random cells in each condition were analyzed with the ImageJ program for the area (A) and perimeter (P); then, we calculated the shape factor (S) with the following equation: S = 4πA/P^2^. S = 1 represents a round shape, and S < 1 represents an elongated shape [23,24].

### 2.8. DCFH-DA Assay

HCT116 (5 × 10^4^ cells/well) were seeded in 96-well plates and incubated at 37 °C with 5% CO_2_ for 24 h. The cells were treated with various conditions. Then, the cells were incubated for 48 h. The media were removed, and 10 μM 2′,7′-Dichlorofluorescin diacetate (DCFH-DA) was added and incubated at 37 °C with 5% CO_2_ for 1 h. The DCF fluorescence intensity was immediately assessed for cellular oxidants at excitation/emission wavelengths of 485/535 nm using a fluorescence microplate reader. A total of 20 μM H_2_O_2_ was used as a positive control [25].

### 2.9. Western Blotting

We put the treated cells’ extracted total protein on ice with RIPA buffer with mixed protease inhibitors (1:10). Cell lysates were centrifuged at 1.2 × 10^4^ g at 4 °C for 20 min. The concentrations of the supernatants were detected with Bradford solution and calculated protein volume. The proteins were separated using 10% sodium dodecyl sulfate–polyacrylamide gel (SDS-PAGE) and transferred to a polyvinyldifluoride (PVDF) membrane. Next, the target proteins were detected with primary antibodies (mouse anti-human E-cadherin, rabbit anti-human TrxR-1) with skimmed milk solution for 12 h at 4 °C. At the end, membranes were washed for 10 min, 3 times with 1× Tris-buffered saline with Tween-20 (TBST), and then were added to the secondary antibodies conjugated with HRP (Abcam, Boston, MA, USA) and developed with Forte Western HRP substrate (Merck, Germany). Finally, the target proteins were detected and calculated with the Chemidoc imaging system. β-actin was used as a standard loading control [26]. Original blots see Appendix A.

### 2.10. Zebrafish Model

The procedures were performed according to the animal health care guidelines. All procedures were approved by the Faculty of Science, Mahidol University Animal Care and Use Committee SCMU-ACUC Review, and received the protocol number MUSC-64-009-558. Zebrafish (*Danio rerio*) were supplied using a GAP-approved aquarium fish farm. The zebrafish were kept at 27 ± 3 °C and a pH of 7.0 ± 1.0, with re-filtered water circulation and a 14/10 h light/dark cycle. Zebrafish were fed two times daily with fleck food supplemented with live brine shrimps (*Artemia salina*). All the zebrafish were quarantined under health monitoring for 1 week before the experiment in a colony tank (20 L) [27].

### 2.11. Xenograft Procedure

The zebrafish were exposed to 10 mg/L dexamethasone (Sigma-Aldrich, USA) for 48 h before the xenograft, and remained exposed to dexamethasone throughout the experiment. A total of 50% of the aquarium water was renewed using 10 mg/L dexamethasone every 48 h. The fish were anesthetized by immersion in 150 mg/L tricaine methanesulfonate (MS-222) (Sigma-Aldrich, USA) for stage III-2 (surgical) anesthesia, which was confirmed by a loss of equilibrium and response of tail-fin pinch; then, the fish were set in the correct position on a wet agarose pad. The treated cancer cells were embedded in ECM gel, and Engelbreth-Holm-Swarm murine sarcoma (Sigma-Aldrich, USA) was injected into the intraperitoneal cavity using a stereoscope with 10,000 cells/20 μL/fish. The xenografted zebrafish were left to recover in the freshwater tank (2 L) for 15 min, and individually reared in the experiment tank (2 L) with an oxygen supply for 7 days [28].

### 2.12. Histology Examination and Immunohistochemistry

The zebrafish were sacrificed with an overdose of MS-222 (300 mg/L) on day 7. The whole bodies of the fish were fixed in 4% paraformaldehyde and decalcified with 10% EDTA at a pH of 7.4 for 1 week. The fish were tissue-processed, paraffin-embedded, and sectioned (6–8 μm) with a longitudinal section. The sections were stained with H&E staining for pathohistological observation. In addition, we detected TrxR-1 expression using rabbit anti-human TrxR-1 (1:100), captured with goat-anti rabbit-HRP (1:10,000); it developed a brown color with DAB substrate and was counterstained with hematoxylin. The image was captured under a light microscope, and we measured the size of the microtumor using the ImageJ program [29].

### 2.13. Statistical Analysis

All the results are presented as the mean ± standard deviation (mean ± SD). Student’s *t*-test and ANOVA were used to compare the significant differences between control and treated cells. Statistical significance was considered at *p* ≤ 0.05 with the SPSS version 23 computer software.

## 3. Results

### 3.1. High Proliferation of Colon Fibroblast Secretes Pro-Inflammatory Cytokines

The first investigation showed an association between the high proliferation of colon fibroblasts and pro-inflammatory cytokine secretion in the fibroblast-cultured media (FCM). The pro-inflammatory cytokines in the media with a high proliferation of colon fibroblasts were investigated. The FCM was collected from the 48 h cultured media of the highly proliferative colon fibroblast. At that time, the confluence of the fibroblast was 95–100% (Figure 1A). The media had a positive variety of cytokines, including growth-related oncogene (GRO)-αβγ, GRO-α, Interleukin (IL)-6 and -8, and Monocyte chemoattractant protein (MCP)-1, -2, and -3. In contrast, the complete media (CM, control) were not detected (Figure 1B,C). These results suggested that the active colon fibroblast secretes the pro-inflammatory cytokines in the fibroblast-cultured media.

### 3.2. Fibroblast-Cultured Media Induce HCT116 Progression

This study aimed to investigate the neighboring fibroblasts’ effects on the progression of colorectal cancer cells. The progression of colorectal cancer cells involved high growth, migration, and invasion capacities in HCT116 cells, a highly aggressive cancer. The MTT assay illustrated cell viability, which showed that FCM induced significant growth (165.96 ± 11.15%) compared to normal conditions (CM) (Figure 2A,B). The wound-healing assay was performed for HCT116 cells that migrated. This result showed that the FCM-treated cells significantly activated HCT116 migration each day (32.22 ± 8.18, 65.92 ± 6.14, and 83.07 ± 7.34 percent of the wound area at 24, 48, and 72 h, respectively). The endpoint (72 h) showed that the FCM-treated cells also significantly increased HCT116 migration compared to the control (Figure 2C,D). The invasive capacity observation found that the FCM-treated cells had a significantly higher number of invasive cells (449 ± 86 cells/field) than the control (130 ± 63 cells/field) (Figure 2E,F). These results suggested that FCM can induce the aggressive progression of HCT116 colorectal cancer cells.

### 3.3. Fibroblast-Cultured Media Induce the EMT Process in HCT116 Cells

The epithelial–mesenchymal transition (EMT) process is a crucial step of the cancer cell in cancer progression, migration, invasion, and metastasis. This study aimed to investigate epithelial markers and the morphological alteration of HCT116 cells after being induced with FCM. Immunofluorescence found that FCM-treated cells showed a loss in the localization of E-cadherin at the junction between cells. In addition, the protein expression according to Western blot analysis also showed a significant reduction in E-cadherin in FCM-treated cells (0.69 ± 0.12) compared with the control (1.00 ± 0.00) (Figure 3A–C). The morphological alterations were also interpreted using shape factors whereby equal to one is a circular or polygonal shape cell, and less than one is an elongated cell, which refers to a mesenchymal appearance. The calculation of the shape factor showed 0.94 ± 0.07 and 0.61 ± 0.23 for control and FCM-treated cells, respectively. FCM-induced HCT116 had an intensely potent ability to alter the morphology from a polygonal shape to an elongated shape (Figure 3D,E). Therefore, the FCM-treated cells showed decreased expression of the epithelial marker, E-cadherin, and there was an increased appearance of mesenchymal cells with elongated cells, indicating the EMT process.

### 3.4. Fibroblast-Cultured Media Increase HCT116 Oxidative Stress Related to Thioredoxin Reductase-1

Oxidative stress is a major factor inducing cancer progression. This study aimed to investigate the roles of highly proliferative fibroblasts in total oxidative stress, and the expression of the oxidative stress-sensitive molecule, TrxR-1, in HCT116 colorectal cancer cells. The cellular oxidative stress of HCT116 cells was detected with a DCFH-DA assay. The results showed that FCM-treated cells had a higher cellular oxidative stress (2.54 ± 0.24) than a normal condition (CM, 1.00 ± 0.00) (Figure 4A). In addition, Western blot analysis was performed to investigate the protein level of TrxR-1. The results showed that the relatives of TrxR-1 are 1.00 ± 0.00 and 1.37 ± 0.15 for control and FCM-treated cells, respectively (Figure 4B,C). These results show that FCM induced TrxR-1 in the high-oxidative-stress condition in HCT116 colorectal cancer cells.

### 3.5. TrxR-1 Relates to Mesenchymal Morphology in Fibroblast-Induced HCT116 Cells

This study focused on the correlations between TrxR-1 expression and cancer cell morphology. The experiment was performed with an immunofluorescent technique to detect the intensity of TrxR-1 expression. The correlation was determined and represented with a dot plot of TrxR-1′s intensity and shape factor. The dot plot illustrated that most control cells showed a round shape and low intensity of TrxR-1; in contrast, the FCM-treated cells showed a high intensity of TrxR-1 in the mesenchymal shape (Figure 5). This result shows that the high expression of TrxR-1 was associated with FCM-induced mesenchymal alteration.

### 3.6. TrxR-1 Indicates HCT116 Dissemination in the Adult Zebrafish Model

This study aimed to investigate the behavior of colorectal cancer cell transplantation in the adult zebrafish model, and its association with TrxR-1 expression. All zebrafish were treated with dexamethasone 48 h before xenotransplantation, and exposure was continued throughout the experiment (7 days). All of the zebrafish survived until the final day. Cancer localization was observed with histological examination. The results showed that FCM induced HCT116 cell scattering in the intraperitoneal cavity and surrounding tissue with high numbers of leukocytes, whereas non-induced cancer cells were still packed inside the ECM gel, with little leukocyte infiltration (Table 1, Figure 6A). The tissues were also investigated for TrxR-1 expression with the immunohistochemistry technique. The result showed that TrxR-1 showed positive results only in the individual cells, in both control and FCM-induced groups. However, the packed tumor showed negative TrxR-1 results (Figure 6B). This shows that FCM-induced cancer dissemination was associated with a TrxR-1-positive status.

## 4. Discussion

The tumor microenvironment, especially that of fibroblasts, has been reported as supportive of cancer cell aggressiveness. The highly proliferative fibroblast was recognized as appearing with typically generated pro-inflammatory cytokines that generally regulate the inflammatory response and tissue-repairing process, and act in an autocrine fashion to activate self-proliferation [30,31,32]. The secretory molecules in the FCM were detected, including GRO-αβγ, GRO-α, interleukin (IL)-6, IL-8, and MCP-1, -2, and -3. Interestingly, the detected secretory molecules were associated with cancer progression, especially massive growth, motility, invasive ability, and the EMT process, which support the result of this study. FCM, which contained the pro-inflammatory cytokines, also induced cellular oxidative stress in HCT116 cells. In previous studies, it was found that oxidative stress was associated with cytoskeleton alteration, cell–cell detachment, and the transcription of matrix metalloproteinases (MMPs) that are involved in cancer cell migration and invasion. [33,34,35]. Moreover, the oxidative stress condition also activated EMT, which is a significant process in metastatic cancer, by the downregulating the epithelial marker, E-cadherin, which is related to upregulating mesenchymal markers such as a morphological change to an elongated shape characteristic of mesenchymal cells [34,36,37]. In addition, FCM-treated cells showed a significant increase in cellular oxidants related to an aggressive appearance. Thus, the pro-inflammatory cytokines of the highly proliferative colon fibroblast in FCM might be essential in increasing cellular oxidative stress associated with HCT116 colorectal cancer cell progression.

Besides the direct effect of fibroblasts on cancer cell alteration, cooperation with leukocytes might support cancer dissemination. Our histological examination illustrated that FCM-treated cells had many scattered microtumors with massive leukocyte infiltration. In contrast, the control group showed an embedded tumor and lower cellular response. Previous studies have suggested that cancer metastasis involves cytokines, chemokines, and growth factor secretions [38,39]. Interestingly, the FCM group featured the aggressive progressions of cancer cells with fibroblast-secretory cytokines, including

GROs; IL-6; IL-8; and MCP-1, -2, and -3. These cytokines also play an essential role in promoting the inflammatory cell response that supports the massive lymphocyte infiltration associated with cancer cell spreading [30,31,32]. In this study, the tumor microenvironment cells, especially fibroblasts and leukocytes, may significantly promote cancer dissemination in the zebrafish models.

TrxR-1 is a cytoplasmic enzyme that responds to oxidative stress by reacting with thioredoxin (Trx) and NADPH to balance the cellular oxidants [14,40]. Thus, increased levels of Trx and TrxR-1 are recognized as oxidative stress markers [16,41]. A previous study suggested that TrxR-1 can promote Murine CT26 colon cancer cell growth [42]. In addition, the overexpression of TrxR-1 has been detected in CRC tissue and CRC cell lines [43]. However, the role of TrxR-1 as an aggressive CRC indicator has not been investigated. Thus, our study aimed to investigate TrxR-1, which might be a biomarker of aggressive CRC progression. FCM was examined to induce the aggressiveness of CRC. FCM-treated cells also increased oxidative stress, and were used to observe the stress marker. Then, TrxR-1 was detected, and it showed overexpression compared to untreated cells. Next, the correlation between TrxR-1 expression and cancer morphology was observed. The results indicated that most of the FCM-treated cells had a high association with the mesenchymal appearance, and were also strongly related to the intensity of TrxR-1. The combination of the morphological change and the overexpression of TrxR-1 might predict CRC progressions in an in vitro model. Moreover, the immunohistochemical examination in the zebrafish model indicated that TrxR-1 was positively associated with individual cancer cells that are highly spreadable. Thus, this study suggests that TrxR-1 has potential use as a novel biomarker for fibroblast-associated aggressive CRC progression in HCT116 cell and zebrafish models. However, this present study only involves work on the cell line and a transplant model. Therefore, further studies must investigate various colorectal cancer states using patient biopsies for precise prediction and clinical application.

## 5. Conclusions

In conclusion, the high proliferation of fibroblast-associated pro-inflammatory cytokine secretions is linked to an increased ability to induce cellular oxidative stress and CRC progression through the EMT process. FCM also caused cancer dissemination in the zebrafish model. Furthermore, TrxR-1 expression was associated with FCM-inducing cancer progression in the HCT116 cancer cell, and cancer transplantation in the zebrafish model. Thus, TrxR-1 is a potential biomarker to indicate fibroblast-associated colorectal cancer progression. Therefore, TrxR-1 has been suggested as a biomarker candidate for colorectal cancer progression and prognostic evaluation in clinical applications.

## Figures and Tables

**Figure 1 cancers-15-00056-f001:**
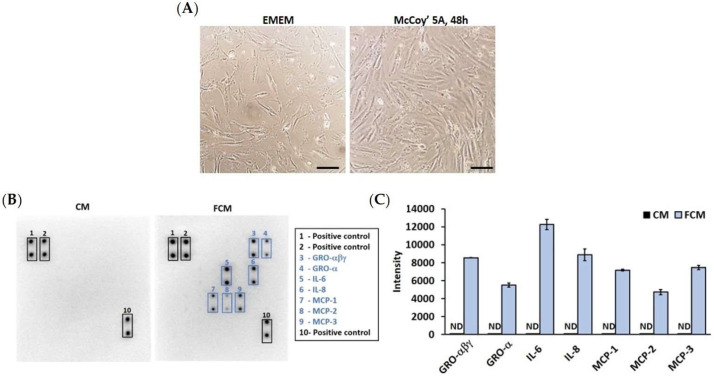
High proliferative colon fibroblast releases pro-inflammatory cytokine in FCM. (**A**) Colon fibroblast (CCD-18co) cultured in Eagle-minimum essential media (EMEM) for approximately 80% confluence. Then, the fibroblast was changed to Mccoy’5A media for 48 h culture. The fibroblast represented approximately 95–100% confluence (magnification 100×). (**B**) Cytokine array was performed to detect the cytokine content in FCM which incubated the active colon fibroblast for 48 h. (**C**) The intensity of pro-inflammatory cytokine of CM and FCM. CM = complete media (control), FCM = fibroblast-cultured media, ND = not detected.

**Figure 2 cancers-15-00056-f002:**
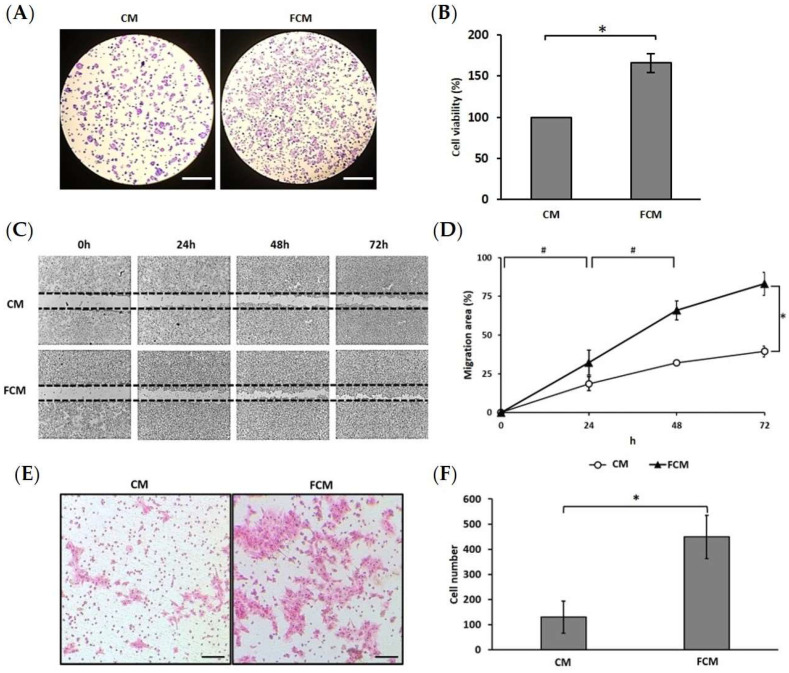
Effect of FCM on cancer progression. (**A**) The cancer cells stained with crystal violet represent the cell occurrence (magnification 40×). (**B**) The bar graph represents HCT116 cell viability by MTT assay with the comparison between complete media (CM), and fibroblast-cultured media (FCM) treated cells. (**C**) Wound healing assay was performed. The cells captured the movement every single day under the light microscope after being treated (magnification 40×). (**D**) Line graph represents the percentages of migration areas in each condition. (**E**) Boyden-chamber assay was performed to investigate the invasion (magnification 100×). (**F**) The graph represents the average number of cell invasion in each field. * Represents the significance of differences between groups at *p* ≤ 0.01, ^#^ represents the significance of differences between time at *p* ≤ 0.01.

**Figure 3 cancers-15-00056-f003:**
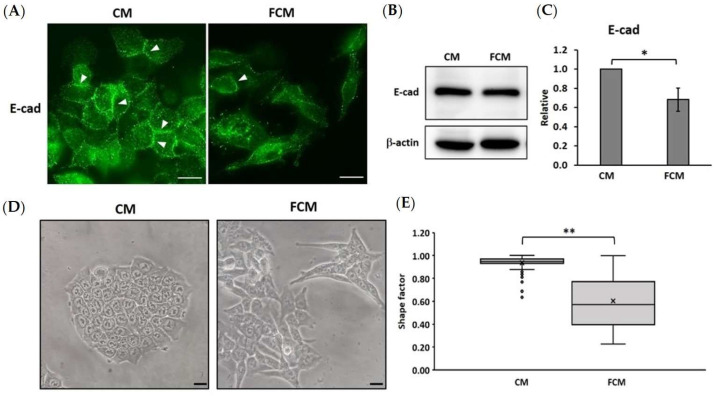
FCM activate EMT process in HCT116 cells. (**A**) Immunofluorescence assay represents a loss of adherent molecule, E-cadherin when treated with FCM. Arrowhead points to the junction between the cancer cells (magnification 400×). (**B**) Western blot analysis represents a decreasing of E-cadherin expression in FCM-treated cell and (**C**) the graph illustrates a relative intensity of E-cadherin protein band with β-actin. (**D**) Cancer cells were treated with FCM for 48 h to investigate the morphology changed under microscope (magnification 400×). (**E**) Morphology appearance was analyzed by shape factor equation. The analyzed data are illustrated with box plot (n = 100). * Represents *p* ≤ 0.05 and ** represents *p* ≤ 0.01 compared between groups. CM = complete media (control), FCM = fibroblast-cultured media, E-cad = E-cadherin.

**Figure 4 cancers-15-00056-f004:**
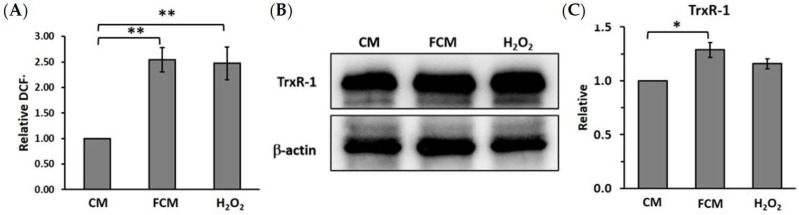
FCM induces cellular oxidant associated with TrxR-1 and cancer progression. (**A**) The relative proportion of cellular oxidant with the treated cell and control. (**B**) TrxR-1 was increased in FCM treated HCT116 that detected with Western blot. β-actin was used as loading control. (**C**) Relative intensity of protein band with β-actin and CM (control). The bar graph is represented with mean ± SD. * represents *p* ≤ 0.05 and ** represents *p* ≤ 0.01 compared between group. CM = complete media (control), FCM = fibroblast-cultured media, and 20 μM H_2_O_2_ as positive stress condition.

**Figure 5 cancers-15-00056-f005:**
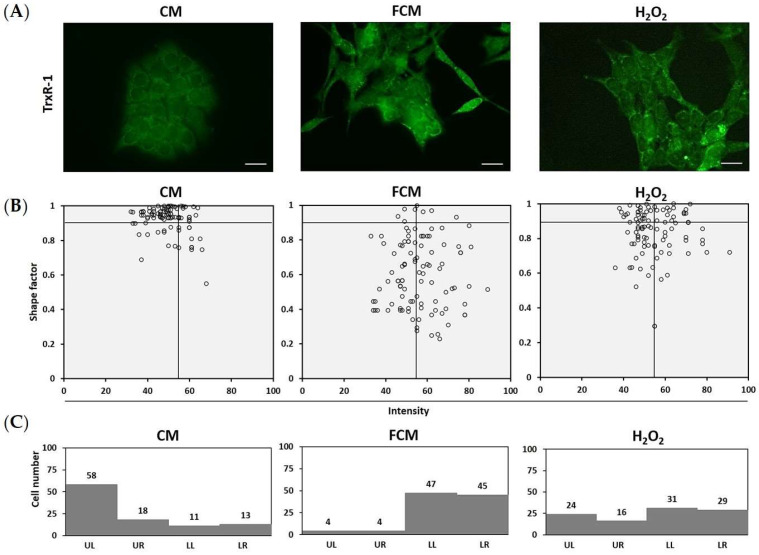
TrxR-1 relates to mesenchymal morphology in fibroblast induced HCT116 cells. (**A**) TrxR-1 expression was detected with immunofluorescence (magnification 400×). The intensity and morphology of HCT116 cells were analyzed with image J software. (**B**) The data represented with dot plot of shape factor and the intensity of TrxR-1 expression (n = 100). (**C**) Bar graphs represent the proportion of cell number in each quadrant (UL = upper-left, UR = upper-right, LL = lower-left and LR = lower-right). CM = complete media (control), FCM = fibroblast-cultured media, and 20 μM H_2_O_2_ as positive stress condition.

**Figure 6 cancers-15-00056-f006:**
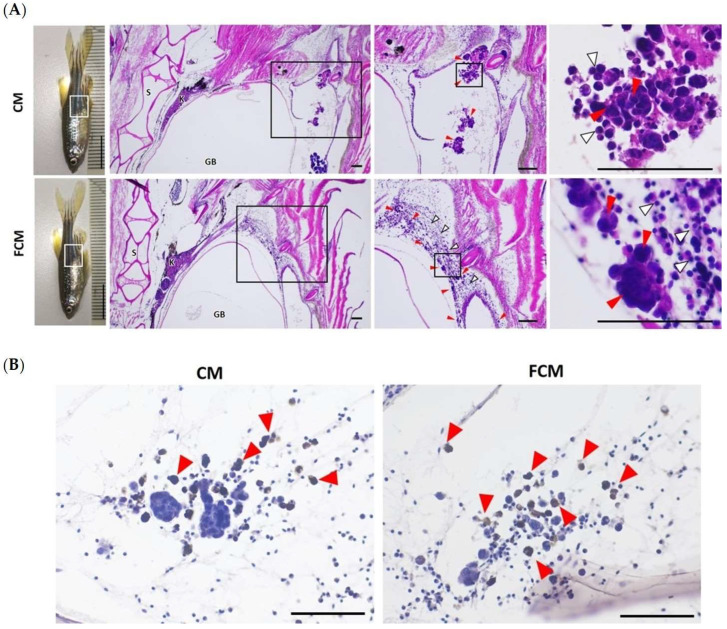
TrxR-1 positive on FCM induced disseminating HCT116 in zebrafish model. (**A**) Pathohistological analysis indicates the microtumors and scattered cancer cells (red arrow) surrounded by leukocytes (white arrow). Magnification 40×, 100×, and 400×, respectively. S = spine, K = kidney, GB = gas bladder, CM = complete media (control), and FCM = fibroblast-cultured media. (**B**) Immunohistochemical analysis represents the positive TrxR-1 (brown color) in the spreading cells (Magnification 400×). CM = complete media (control), and FCM = fibroblast-cultured media.

**Table 1 cancers-15-00056-t001:** Histological analysis of microtumor and inflammatory cell response.

	Number of Microtumor/HPF (n = 6)	Microtumor Size (μm^2^)	Number of Lymphocyte/HPF
CM	4.33 ± 1.51 (3–7)	2870.35 ± 1197.59 (426.66–9887.63)	20.83 ± 12.38 (4–40)
FCM	25.00 ± 13.18 * (14–47)	198.71 ± 100.83 * (30.31–1223.06)	169.00 ± 56.16 * (73–211)

Values are mean ± SD (range). * represents *p* ≤ 0.01 compared between group (n = 6). HPF = high-power field, CM = complete media (control), FCM = fibroblast-cultured media.

## Data Availability

Data sharing not applicable.

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
