# Peer review of "Thioredoxin Reductase-1 as a Potential Biomarker in Fibroblast-Associated HCT116 Cancer Cell Progression and Dissemination in a Zebrafish Model"

_cancers, 2022, doi:10.3390/cancers15010056_

Round 1

Reviewer 1 Report

The work is average but may be improved by the inclusion of the following suggestions.

1.      English should be improved throughout the manuscript.

2.      Quantitative information should be provided in the abstract.

3.      The concussion should be concise and to the points indicating the application of the work.

4.      The novelty of the work should be established.

5.      Please write one paragraph in the introduction on cancer, in general, and you can consult the following articles to make this manuscript more useful to the readers.

Future Med. Chem., 5: 135-146 (2013).; Future Med. Chem., 5, 961-978 (2013).; Med. Chem., 9, 11-21 (2013).; RSC Adv., 4, 29629 - 29641 (2014).

6.      Please write one paragraph in the introduction on chemotherapy, in general, and you can consult the following articles to make this manuscript more useful to the readers.

Can Ther., 8, 6-14 (2011).RSC Adv., 9, 15357-15369 (2019).;  Cur. Pharm. Anal., 1, 109-125 (2005).

8.  Please provide error graphs in the figure; where are required.

9.      Please improve the quality of the Figures.

10.  Please compare your results with previous studies and mention clearly how your work is important in comparison to already been reported.

Reviewer 2 Report

 Cancer manuscript review 11-13-2022

The main question addressed by the researchers is addressed in the abstract and is as follows:

The aim of the study is to investigate the TRxR-1 expression as an indication of colon fibroblast inducing colorectal cancer progression and metastasis.

There is a pictorial graphical abstract which displays the theoretical framework that encompasses the study nicely.

The topic is very original in that there is a paucity of literature that studies all the indicators looked at in this study together that contribute to high proliferative colon fibrosis specifically to colon cancer.

The study is most definitely relevant as it is a known fact that the five year survival rate of patients with metastatic colorectal cancer is quite grim. So it stands to reason that any research pertaining to early progression via new biomarkers is always pertinent and of vital interest to physicians and medical researchers.

This study adds numerous points of research for further study pertaining to all of the potential indicators of colonic cancer progression that contribute to the increases of TRxR-1 specifically to colon cancer. It paves the way for further research in human models.

The study its model is designed from previous research that looks at segments of this purported model, and this is portrayed in the introduction.

The conclusion are in line with the justification of the findings. I would though add in the conclusion that further research is needed on human models to justify their conclusions.

Likewise I would add this to the abstract and summary sections.

Also needed is a section called areas of further research as pertains to these findings.

The references are appropriate as they validate the different segments of the research.

The tables and figures are excellent, as they validate the findings in   and tabular form.

Reviewer 3 Report

In this paper, the authors designed the tumor environment, especially fibroblast, strongly promotes colorectal cancer progression due to the accumulation of reactive oxygen species (ROS), leading to oxidative stress. Oxidative stress is associated with the expression of TrxR-1, which is an oxidative stress sensitivity molecule. This study focused on the investigating of TrxR-1 expression as an indication of colon fibroblast-inducing colorectal cancer progression and metastasis. Authors found that the high proliferative fibroblast-cultured media contained pro-inflammatory cytokines, which have a high ability to influence CRC progression. In addition, TrxR-1 level was associated with cellular oxidant enchantment, indicating FCM inducing the EMT process and cancer dissemination in both HCT116 colorectal cancer cell and zebrafish models. Furthermore, TrxR-1 could be applied as a biomarker for colorectal cancer progression and prognostic evaluation. However, numerous mistakes would misunderstand and confuse readers. Several typo-error should be re-checked and revised. Since, I recommended accepted this paper after major revision. Some corrections and suggestions were listed below:

(1)   In Figure 5, the author should describe the title more clearly. For example, the author should mark the Figures with (A), (B), and (C) as same as Figure 1, and describe the title respectively.

(2)   In the line 45, 54, 63, 68 …, author should remove the unnecessary space of the cited reference. For example, the line 45 should be revised from “…in the mortality rate [3, 4].” to “…in the mortality rate [3,4].”.

(3)   In Figure 1, author should revise the mark of Figure 1C in the title. The line 232 should be revised from “B) The intensity of pro-inflammatory…” to “C) The intensity of pro-inflammatory…”

(4)   The name of the used cell line should be consistent. For example, “HCT116” in the title; “HCT-116” in the Abstract; and “HCT 116” in line 301.

Round 2

Reviewer 1 Report

The work is average but may be improved by the inclusion of the following suggestions.

1.      English should be improved throughout the manuscript.

2.      Quantitative information should be provided in the abstract.

3.      The concussion should be concise and to the points indicating the application of the work.

4.      The novelty of the work should be established.

5.      Please write one paragraph in the introduction on cancer, in general, and you can consult the following articles to make this manuscript more useful to the readers.

Future Med. Chem., 5: 135-146 (2013).; Future Med. Chem., 5, 961-978 (2013).; Med. Chem., 9, 11-21 (2013).

6.      Please write one paragraph in the introduction on chemotherapy, in general, and you can consult the following articles to make this manuscript more useful to the readers.

Can Ther., 8, 6-14 (2011), RSC Adv., 9, 15357-15369 (2019), RSC Adv., 4, 29629 - 29641 (2014).

7.      Please write one paragraph in the introduction on biomarkers, in general, and you can consult the following articles to make this manuscript more useful to the readers.

J. Sep. Sci., 39, 69-82 (2016), Recent Pat. Biomark., 5, 81-92 (2015), Recent Patents on Biomarkers, 1, 89-97 (2011).

8.  Please provide error graphs in the figure; where are required.

9.      Please improve the quality of the Figures.

10.  Please compare your results with previous studies and mention clearly how your work is important in comparison to already been reported.

Reviewer 2 Report

The article presentation is much improved. The topic is also very pertinent. I would accept in present form for publication. Thank-you for the honor and privilege in allowing me to review your fine work. The best of luck in your future endeavors.

Author Response

We appreciate your precious time reviewing our paper and providing valuable comments.

Reviewer 3 Report

它可以在現在發布。 

Author Response

(The authors gave the same response as above.)

Round 3

Reviewer 1 Report

Accepted